# Ultra-high-resolution observations of persistent null-point reconnection in the solar corona

X. Cheng [1,2,3] ✉, E. R. Priest[4], H. T. Li [1,3], J. Chen[1,3], G. Aulanier [5,6], L. P. Chitta [2], Y. L. Wang [1,3], H. Peter [2], X. S. Zhu[7], C. Xing[1,5], M. D. Ding [1,3], S. K. Solanki [2], D. Berghmans[8], L. Teriaca [2], R. Aznar Cuadrado [2], A. N. Zhukov [8,9], Y. Guo [1,3], D. Long [10], L. Harra[11,12], P. J. Smith [10], L. Rodriguez[8], C. Verbeeck [8], K. Barczynski [12] & S. Parenti[13]

Magnetic reconnection is a key mechanism involved in solar eruptions and is also a prime possibility to heat the low corona to millions of degrees. Here, we present ultra-high-resolution extreme ultraviolet observations of persistent null-point reconnection in the corona at a scale of about 390 km over one hour observations of the Extreme-Ultraviolet Imager on board Solar Orbiter spacecraft. The observations show formation of a null-point configuration above a minor positive polarity embedded within a region of dominant negative polarity near a sunspot. The gentle phase of the persistent null-point reconnection is evidenced by sustained point-like high-temperature plasma (about 10 MK) near the null-point and constant outflow blobs not only along the outer spine but also along the fan surface. The blobs appear at a higher frequency than previously observed with an average velocity of about 80 km s⁻¹ and lifetimes of about 40 s. The null-point reconnection also occurs explosively but only for 4 minutes, its coupling with a mini-filament eruption generates a spiral jet. These results suggest that magnetic reconnection, at previously unresolved scales, proceeds continually in a gentle and/or explosive way to persistently transfer mass and energy to the overlying corona.

Magnetic reconnection due to changes in connectivity of magnetic field lines is a fundamental energy release mechanism in plasmas[1]. During large-scale solar eruptions, reconnection is thought to take place in an elongated current sheet that connects the erupting coronal mass ejections (CMEs) and flare loops[2–4]. In this scenario, reconnection can transfer flux from a coronal arcade into the twisted flux rope, which is then added to the pre-eruptive configuration, helping to drive the fast formation and eruption of CMEs[5–7]. At the same time, it can

[1]School of Astronomy and Space Science, Nanjing University, 210093 Nanjing, China. [2]Max Planck Institute for Solar System Research, 37077 Göttingen, Germany. [3]Key Laboratory of Modern Astronomy and Astrophysics (Nanjing University), Ministry of Education, 210093 Nanjing, China. [4]School of Mathematics and Statistics, University of St. Andrews, Fife KY16 9SS Scotland, UK. [5]Sorbonne Université, Observatoire de Paris - PSL, École Polytechnique, IP Paris, CNRS, Laboratory for Plasma Physics (LPP), 4 place Jussieu, 75005 Paris, France. [6]Rosseland Centre for Solar Physics, Institute for Theoretical Astrophysics, Universitetet i Oslo, P.O. Box 1029, Blindern, 0315 Oslo, Norway. [7]State Key Laboratory of Space Weather, National Space Science Center, Chinese Academy of Sciences, Beijing, China. [8]Solar-Terrestrial Centre of Excellence - SIDC, Royal Observatory of Belgium, Ringlaan -3- Av. Circulaire, 1180 Brussels, Belgium. [9]Skobeltsyn Institute of Nuclear Physics, Moscow State University, 119992 Moscow, Russia. [10]Mullard Space Science Laboratory, University College London, Holmbury St. Mary, Dorking, Surrey RH5 6NT, UK. [11]PMOD/WRC, Dorfstrasse 33, CH-7260 Davos Dorf, Switzerland. [12]ETH-Zürich, Wolfang-Pauli-Strasse 27, HIT J 22.4, 8093 Zürich, Switzerland. [13]Institut d'Astrophysique Spatiale, Université Paris-Saclay, 91405 Orsay Cedex, France. ✉e-mail: xincheng@nju.edu.cn

efficiently accelerate particles in the corona, which stream down to and heat the lower atmosphere, giving rise to chromospheric flare emission[8,9]. Such a picture has been extensively studied in past decades primarily through remote sensing spectroscopic and imaging observations. Some significant and critical features predicted by the model were identified observationally, including reconnection inflows and downflows[10–12], outflow-driven termination shock[13], and rapid change of magnetic flux connectivity[14,15].

Magnetic reconnection essentially takes place at small scales down to tens of metres in the corona. The large-scale current sheet during solar eruptions is conjectured to be composed of fragmented current elements (or magnetic islands) of different scales, likely arising from tearing mode instability and turbulence[16–20]. Intermittent sunward outflow jets with a wide velocity distribution provide a strong indicator of fragment of the large-scale current sheet[4]. Similar processes were also believed to occur during small-scale reconnection events in the lower atmosphere[21–23]. Using high-resolution Hα images from the ground-based New Vacuum Solar Telescope (NVST)[24], it was clearly observed that plasmoids are expelled out of small-scale reconnection regions intermittently[25,26].

Magnetic reconnection is also a promising candidate for releasing the energy to heat the corona to millions of degrees[27,28]. Taking advantage of extreme-ultraviolet (EUV) imaging data from the High-resolution Coronal Imager (Hi-C), which ideally is able to resolve scales on the order of 150 km, ref. 29 provided evidence for reconnection

between braided magnetic threads and corresponding heating[30,31]. Using the Imaging Magnetograph eXperiment (IMaX) instrument[32] on board SUNRISE[33], with a spatial resolution of ~80 km, ref. 34 found that active-region coronal loops could be located in regions where small-scale opposite polarities cancel with the dominant polarity and inverse Y-shaped jets are frequently ejected. The two features strongly indicate that cancellation-driven small-scale reconnection plays a vital role in transferring energy and mass into the coronal loops[35,36].

Here, we report a small-scale null-point reconnection event observed by the EUV High-Resolution Imager (HRI$_{EUV}$) 174 Å of the Extreme-Ultraviolet Imager (EUI)[37] onboard Solar Orbiter (SolO)[38] on 3 March 2022 as located at a distance of 0.55 AU from the Sun. High spatiotemporal resolution images of the HRI$_{EUV}$ revealed that the reconnection driven by a moving magnetic feature takes place continuously at the null-point, at previously unresolved scales, over the period (one hour) of the EUI observation.

## Results
### Persistent null reconnection
In the high-resolution HRI$_{EUV}$ 174 Å images, a point-like brightening with a spatial scale of ~390 km (two pixels) is visible throughout the sequence (Figs. 1 and 2a–f). The Helioseismic and Magnetic Imager (HMI)[39] line-of-sight (LOS) magnetograms show that the point-like brightening is located above a minor isolated positive polarity embedded within the main negative polarity (Fig. 2s). These features

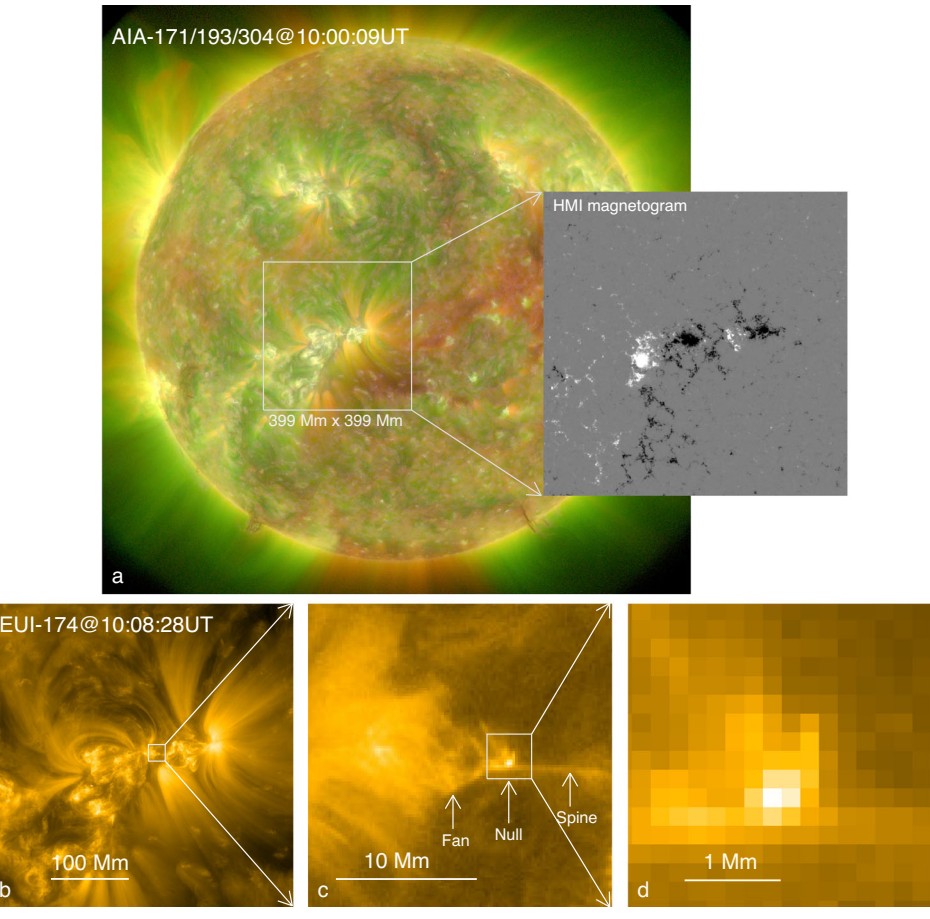

**Fig. 1 | Overview of observations. a** Composite of the AIA 171 Å, 193 Å, and 304 Å passbands showing the full disk EUV image of the Sun at 10:00 UT on 2022 March 3 overlaid by the simultaneous HMI LOS magnetogram for the same FOV of the EUI/HRI (white box) with the positive (negative) polarity in white (black). **b** The EUI/HRI$_{EUV}$ 174 Å full FOV image displaying the fine structure of NOAA 12957. **c** Zoom-in of part of HRI$_{EUV}$ FOV (white box in **b**) showing a fan-like bright structure suggesting

a null-point and fan-spine configuration as pointed out by three arrows and confirmed by a magnetic extrapolation in Fig. 3. **d** Zoom-in of the point-like brightening (white box in **c**) indicating the spatial scale of heated plasma associated with the null reconnection. Note that, the EUI and AIA images are not exactly co-aligned because of the distortion caused by the separation of SolO from the Earth by 7 degrees.

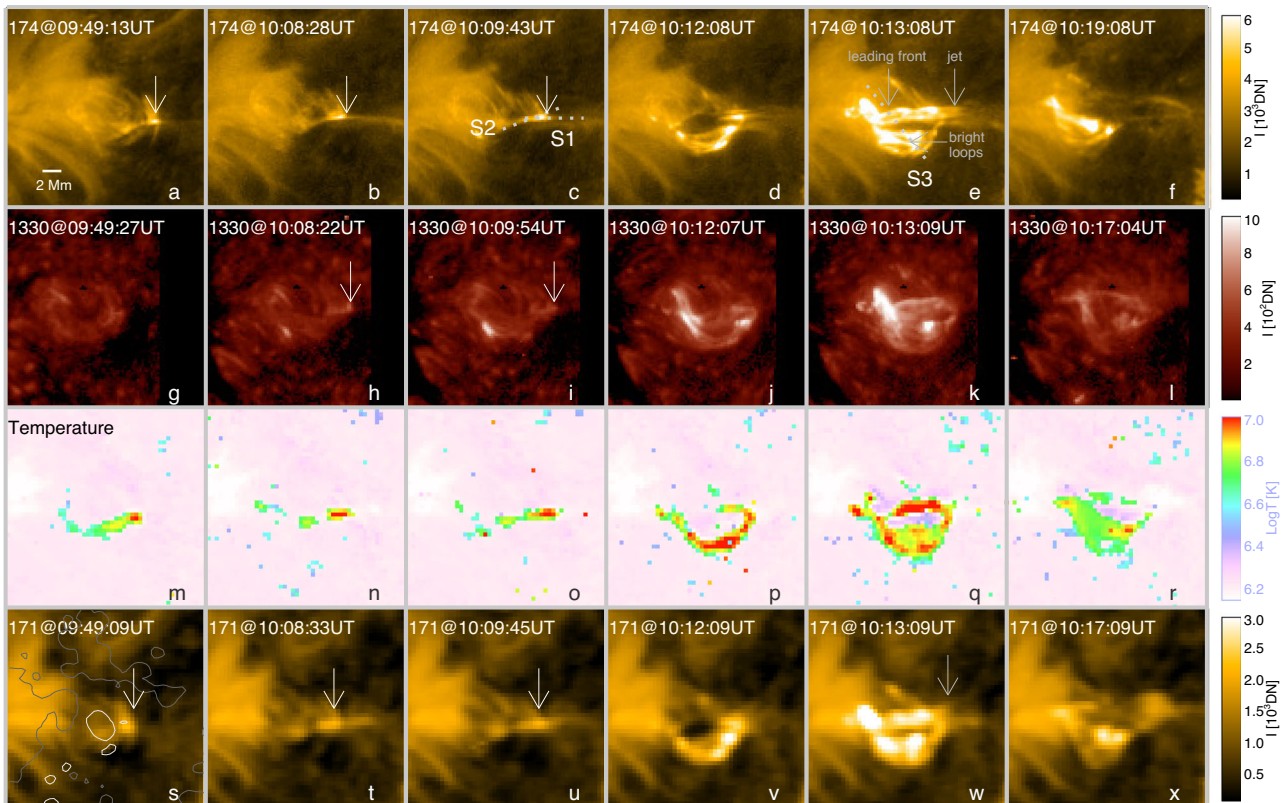

**Fig. 2 | Signatures and thermal structuring of null-point reconnection. a–f** Time sequence of EUI/HRI$_{EUV}$ 174 Å images indicating persistent reconnection at the null-point (**a–c**) and the eruption of a spiral jet (**d–f**). The two slits (S1 and S2) with a width of one pixel are used for making the distance-time plots as shown in Fig. 4a, b; the S3 slit is used to create Fig. 4e. All times have been corrected to that at the Earth. **g–r** Same as **a–f** but for the IRIS 1330 Å slit-jaw images (**g–l**) and DEM-weighted average temperature maps (**m–r**). **s–x** The AIA 171 Å images of persistent reconnection at the null-point (**s–u**) and spiral jet (**v–x**), the contours (±100G) of the HMI LOS magnetogram at the same time are overlaid in **s** with white (grey) indicating positive (negative) polarity. The point-like brightening indicating the null point is pointed out by the white arrows in **a–c**, **h–i**, **s–u**. The spiral jet, leading front of the erupting filament, and induced bright loops are indicated by the grey arrows in **e** and **w**.

suggest that the magnetic structure consists of a magnetic dome enclosing the flux that connects to the isolated positive polarity and separating that flux from the surrounding negative polarity. This is confirmed by extrapolating the three-dimensional (3D) coronal potential field structure from an observed photospheric magnetogram (Fig. 3), which clearly shows that there is indeed a dome containing a 3D null-point and representing the fan separatrix surface of magnetic field lines that spread out from the null. Justification for the potential extrapolation is given in Section Methods. In addition, two isolated spine field lines approach the null from above and below. The cospatiality between the point-like brightening and the null-point strongly indicates that the reconnection takes place near the null-point at the intersection of the spine and fan as in the theory of ref. 40. Such a structure corresponding to the field around a 3D magnetic null-point has also been suggested for a UV burst close to the photosphere[41,42], for jets at the base of coronal plumes or equatorial coronal-holes[43,44] and for a flare with circular ribbons[45,46].

In our case, the point-like brightening also occasionally shows up in the Interface Region Imaging Spectrograph (IRIS)[47] 1330 Å images in spite of being more tenuous (Fig. 2g–l), implying that this localised EUV brightening sometimes produces a weak counterpart at the lower atmosphere, related to the low height (about 2 Mm) of the null-point (Fig. 3c). Compared with HRI$_{EUV}$ 174 Å, the brightening is recognisable as well at the Atmospheric Imaging Assembly (AIA)[48] 171 Å passband but with less detail. The differential emission measure (DEM) analyses based on six co-aligned AIA EUV passbands show that the emissions near the null-point are distributed over a very wide temperature range

from 0.5 MK to 20 MK (Supplementary Fig. 1) with a DEM-weighted average temperature of ~10 MK (Fig. 2m–r and Supplementary Fig. 1).

An important finding is that the high-temperature near the null-point is maintained for almost the entire observing period of nearly 1 hour, which suggests continuous magnetic reconnection at that location. From the attached HRI$_{EUV}$ 174 Å movie (Supplementary Video 1), one can clearly see that some plasma blobs are continuously expelled from the null-point and propagate along the fan and spine (Fig. 2a–f). For a few blobs also captured by the AIA (Supplementary Video 2), we also calculated their average temperature, which is found to be ~2 MK (Supplementary Fig. 2), lower than that of the null-point associated brightening. These persistent blobs were also detected to come out at the jet base and then move upward in the polar coronal hole[49]. We made two distance-time plots to track the trajectories of these blobs. Figure 4a shows that the blobs move outward along the spine (slit S1 in Fig. 2c). Figure 4b indicates that the blobs move along the inclined direction (slit S2), from which we detected bidirectional motions. After identifying the positions of the fast-moving blobs manually in distance-time plots (dotted lines in Fig. 4a, b), we estimated their linear velocities and lifetimes, the histograms of which are displayed in Fig. 4c, d, respectively. We find that the velocity of the blobs ranges from 30 km s$^{-1}$ to 210 km s$^{-1}$ with an average (median) value of 78 (68) km s$^{-1}$. Given that the blobs may propagate along the outer spine as shown in Fig. 3, the real average velocity is corrected to be around 200 km s$^{-1}$. The lifetime varies from 5 s (i.e., limited by cadence of HRI observations) to 105 s with an average (median) value of 40 (38) s.

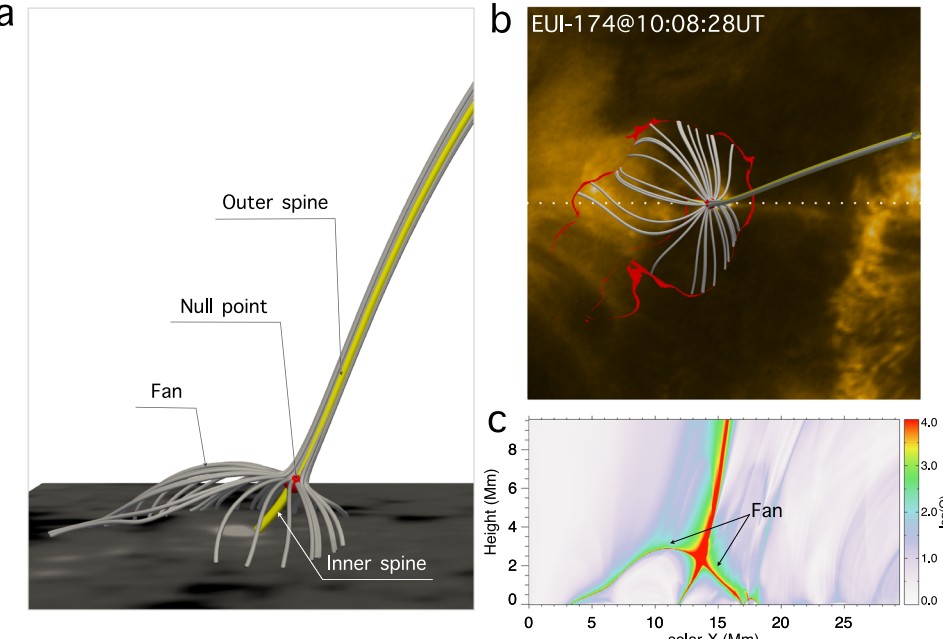

**Fig. 3 | 3D potential configuration of the null-point and associated fan-spine.** **a** 3D magnetic field lines near the fan surface (grey) and the inner and outer spine lines (yellow) with a null-point (red) located at the crosspoint between the spine and fan surface. The bottom boundary shows the radial component of the HMI vector magnetogram with white (black) denoting positive (negative) polarity. **b** Top view of the 3D null-point configuration. The bottom boundary is the simultaneous HRI$_{EUV}$ 174 Å image overlaid by high log $Q$ (>4) regions (red) corresponding to imprints of the fan surface at the photosphere. **c** The distribution of log $Q$ at the x–z plane as indicated by the dotted line in **b** showing the cross-section of the null-point configuration.

## Spiral jet eruption

From about 10:06 UT, a mini-filament near the null-point started to rise slowly. This slow-rise lasted for ~5 min, very similar to the slow-rise usually appearing prior to the main eruption of CMEs[50]. At about 10:11 UT, the mini-filament commenced a faster eruption, with its footpoints quickly drifting toward the northeast (Fig. 2d–f and v–x). At the same time, the erupting flux that consisted of dark and bright threads presented a spiral motion (Fig. 2e, k), accompanied by the plasma ejection toward the west, indicating the release of magnetic twist into the higher corona and heating of part of filament threads. More details can be found in Supplementary Video 1. Moreover, we also observed a pair of localised brightenings underneath the erupting mini-filament with the left one following the drifting mini-filament footpoint. Subsequently, a group of short bright loops with a temperature of about 10 MK appeared (Fig. 2q) and connected the two localised brightenings (Fig. 2e). These processes with more details as revealed here confirm the scenario proposed for blowout jets caused by mini-filament eruptions in previous lower resolution data[51–54].

Compared with previous studies, the current observations show an additional characteristic, i.e., the heating and lateral propagation of the mini-filament leading front when interacting with the fan surface. In the rise phase (Phase I in Fig. 4e), the velocity of the mini-filament leading front was ~10 km s$^{-1}$. After entering Phase II, the left leg of the mini-filament quickly moved laterally with an average velocity of ~150 km s$^{-1}$. It may correspond to the propagation velocity of the fan reconnection as the erupting mini-filament broke through the fan surface. The fan reconnection not only heats the filament threads but also transfers magnetic twist to the overlying field as suggested by ref. 55. Thanks to the better spatial resolution and temporal cadence, these detailed dynamical processes for the spiral jet were more clearly captured by the HRI$_{EUV}$ than by the AIA. In addition, no evidence was found to support that the preceding null-point reconnection plays a role in triggering the mini-filament eruption and causes the spiral jet as argued previously[43,44,56,57] given the null-point reconnection remained sustained after the eruption (Fig. 4a, b).

## Light curves of null reconnection and spiral jet

Although reconnection near the null-point was persistent, it seemed to vary in time. This is indicated by a number of spikes with different amplitudes appearing in the time profile of the HRI$_{EUV}$ 174 Å integrated intensity (the black curve in Fig. 5b), which was derived by integrating a small region only including the null-point (Box 1 in Fig. 5a). In contrast, only some big spikes are detectable in the AIA 171 Å integrated intensity curve of the same region (the grey curve in Fig. 5b). The two spikes during the time period of 10:12–10:14 UT even appear to be higher than that in the HRI$_{EUV}$ 174 Å curve. This might be caused by a temperature effect. In general, hotter plasma has the tendency to show less variability, and the AIA 171 Å passband samples slightly cooler plasma (mainly Fe IX emission) than the HRI$_{EUV}$ 174 Å passband (mostly Fe X). Moreover, we also calculated the 174 Å integrated intensity, as well as the AIA 94 Å, 171 Å, and 304 Å ones, over the whole fan and spine structure (Box 0). It is found that only a few large spikes were able to be identified in the AIA 171 Å and 304 Å light curves. For the AIA 94 Å light curve, no such spikes can be detected. This discrepancy demonstrates a need for EUV imaging with a spatial resolution better than an arcsecond and a time cadence higher than 10 s to resolve the fine structures of small-scale events in the corona and disclose their hidden dynamics[58].

Figure 5c shows the temporal variations of the HRI$_{EUV}$ 174 Å intensities at three regions during the jet eruption (Box 2-4 in Fig. 5a). One can see that the brightenings first appeared at Box 2 and 3, even though not strong. Starting with 10:12 UT, the intensities at all three regions quickly increased and lasted for at least 5 minutes, even for almost 10 minutes at Box 2. Comparing to Box 2 and 3, the peak of the intensity at Box 4 was delayed by 1.5 min, which may be caused by the external fan reconnection commencing after the inner reconnection below the mini-filament. These features again imply that the erupting mini-filament have experienced multiple reconnection processes as it broke through the null-point-associated fan surface.

Note that the spikes of the HRI$_{EUV}$ 174 Å emission from the null-point reconnection seem to be invisible after the jet eruption (Fig. 5b;

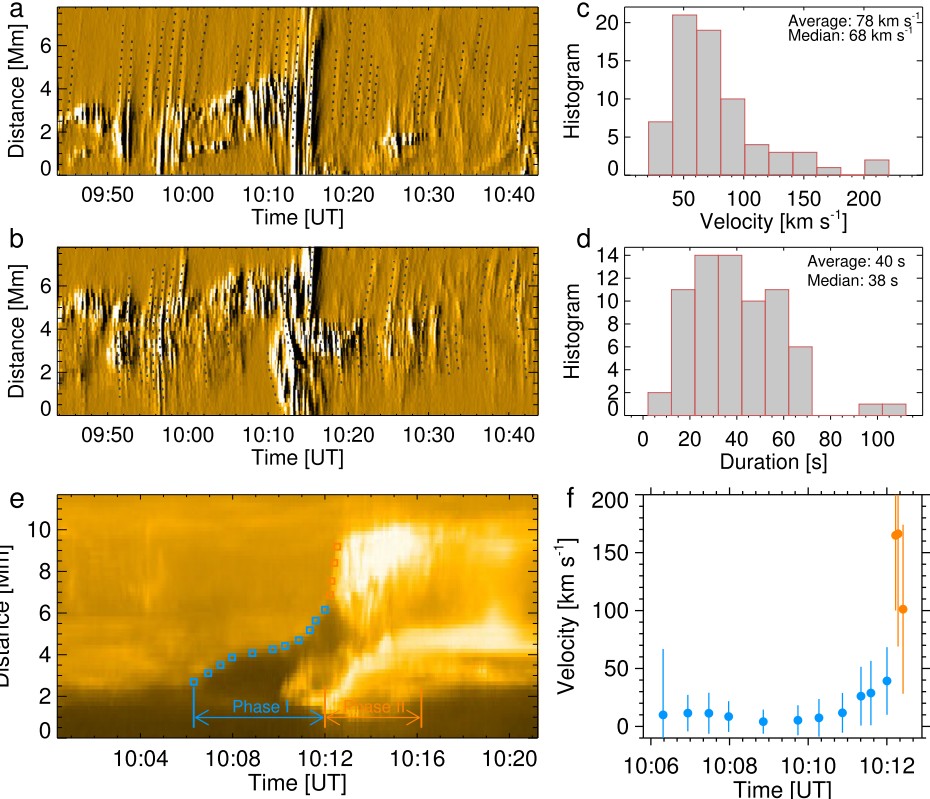

**Fig. 4 | Dynamics associated with observed null-point reconnection. a** Distance-time plot of the HRI$_{EUV}$ 174 Å running-difference images showing the outflow blobs along the spine indicated by the slit S1 in Fig. 2c. The velocities were calculated by linear fitting of the height-time measurements as shown by dotted lines. **b** Same as **a** but for the inclined slit S2 displaying the outflow blobs along the fan surface and the orientation slightly deviating from the outer spine. **c, d** Histograms of the blob velocity and lifetime, respectively. **e** Same as **a** but for the HRI$_{EUV}$ 174 Å original images along the slit S3 in Fig. 2e showing the early rise and heating of the mini-filament. The squares in blue denote distance-time measurements of the mini-filament leading front (phase I); the squares in orange represent the lateral propagation of the fan reconnection (phase II). **f** Temporal evolutions of the velocities of the mini-filament leading front and fan reconnection front (obtained from locations indicated in **e**). The vertical bars represent the errors that are from the uncertainties in distance measurement (about four pixels).

10:16 UT), but the fast-moving plasma blobs are still detectable as shown in Fig. 4a and b. It implies that, even though the heating may be weakened, the null-point reconnection is still ongoing. This is also indicated by the long-term stability of the null-point and persistent plasma heating after the jet eruption (Supplementary Videos 3 and 4).

### Driver of null reconnection and spiral jet

To explore the reasons for the persistent null reconnection and spiral jet eruption, we investigated the long-term evolution of the HMI LOS and vector magnetograms. It was found that the reconnection and jet were closely related to the movement of the minor positive polarity. Highly resembling a typical moving magnetic feature[59–61], it rapidly emerged within the negative polarity penumbra and was then carried outward by the moat flow of the sunspot. We calculated the total flux of the minor positive polarity over the region where the magnetic field strength is larger than 10 G (i.e., above the typical noise level in the HMI line of sight magnetograms). Moreover, we also estimated the height of the null-point during the entire lifetime of the minor positive polarity. Figure 6a shows that the null-point quickly ascends (descends) along with the emergence (submergence) of the minor positive polarity before 07:00 UT (after 11:00 UT). During the time period of 07:00–11:00 UT, even though the height of the null-point only slightly declines, the total flux varies violently, which mainly originates from the interaction of the fast emerging/moving minor positive polarity with the nearby moss region. Meanwhile, the fast movement may also provide a direct driver for continuous reconnection at the null-point.

Furthermore, during the early emergence phase, the flux carried a certain magnetic twist as deduced from the appearance of bald patches (BPs), which represent unusual sections of the photospheric polarity inversion line (PIL) where the twisted field touches the photosphere tangentially[62]. Such twisted field over the PIL often gives rise to filaments or mini-filaments at their dips and represents locations where magnetic free energy is stored. Figure 6b shows that the BPs (red curves) are mainly distributed along the south part of the PIL (blue curves). As time lapsed, the BPs concentrated toward the south. They did not start to disappear until after 11:00 UT. The deduction of magnetic twist is further confirmed by the appearance of a mini-filament that is almost co-spatial with the BPs, in particular during the time period before the jet eruption (Fig. 6c). Moreover, the mini-filament also displays a blueshift feature (Fig. 6c), which suggests that the mini-filament was ascending in height and then erupted to cause the jet once destabilised.

## Discussion

Thanks to high spatio-temporal resolution data of the EUI/HRI$_{EUV}$ recorded during the first orbit of the nominal mission phase of SolO, we have observed reconnection at a small-scale coronal null-point that was formed above a minor positive polarity embedded within the dominant negative polarity of a sunspot and its moat. The important finding is that the magnetic reconnection occurred at the null-point continuously during the entire EUI observation of 1 hour, which is evidenced by persistent point-like high-temperature (about 10 MK) plasma surrounding the null-point and fast-moving plasma blobs along the spine and fan associated with the null-point. This detailed dynamics pertinent to the null-point configuration at the small scale of about 390 km as revealed by the HRI observations could hardly have

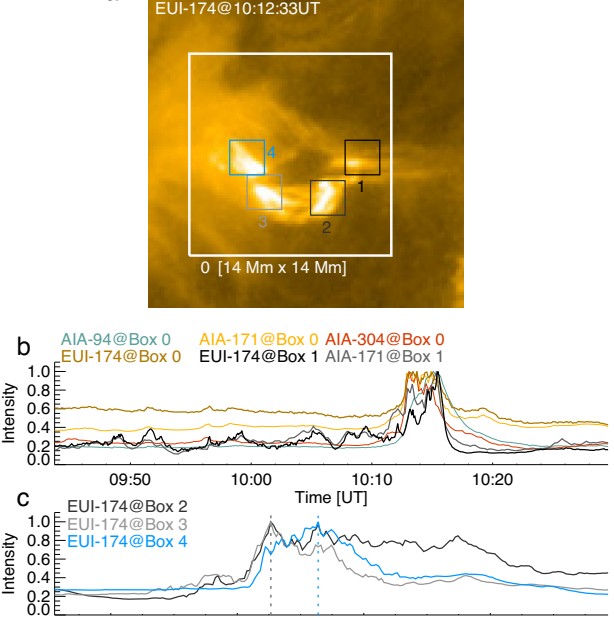

**Fig. 5 | Light curves of null-point reconnection. a** The EUI/HRI$_{EUV}$ 174 Å image at 10:12:33 UT showing the spiral jet caused by the erupting mini-filament. Box 0 covers the area of the whole null-point and fan-spine structure, whereas Box 2–4 outline three regions that brighten. **b** Temporal evolutions of the HRI$_{EUV}$ 174 Å (gold), AIA 94 Å (cyan), 171 Å (yellow), 304 Å (red) integrated intensities within Box 0 whose location is indicated in **a**. The HRI$_{EUV}$ 174 Å (black curve) and AIA 171 Å (grey curve) intensities within Box 1 are overplotted. **c** Temporal evolutions of the HRI$_{EUV}$ 174 Å integrated intensities within Box 2–4.

been detected in previous observations with lower spatial resolution. The origin and topology of the null-point reconnection are summarised in Fig. 7a, b, respectively. Combining it with extrapolated 3D magnetic field configuration, we suggest that the reconnection at the null-point is ceaselessly transferring mass and energy to the overlying corona along the field lines around the outer spine.

The null-point and fan-spine configuration have been observed in flares[44–46,63,64] and at the base of coronal plumes or equatorial coronal-holes[43,44]. In previous studies, although the null-point reconnection was observed to occur repetitively sometimes, the highest occurrence rate was found to be only about once every three to five minutes[44]. Using the EUI/HRI$_{EUV}$ data, it was revealed that the null-point reconnection at scales previously unresolved proceeds almost continuously with hot blobs expelled much more frequently. After carefully examining the time sequence of HMI magnetograms, we suggest that various motions at the photosphere, such as the fast flux movement observed here, may provide a direct driver for the persistence of magnetic reconnection.

The finding of persistent minor null-point reconnection sheds an important light on the solution of the coronal heating problem. As revealed by SUNRISE II, minor small-scale opposite-polarity fluxes are prevalent at the periphery of the penumbra in the moat around a sunspot[34]. In quiet-Sun regions, opposite-polarity fluxes frequently appear within dominant flux concentrations although they may be short-lived[65,66]. We tentatively calculated the topology of the potential field over a larger quiet-Sun region nearby the null-point studied currently and found abundant low-lying small-scale null-points, consistent with previous explorations[67–69]. Our observations thus support to find even smaller and more frequent null-point reconnection events, in particular over the quiet-Sun region, hopefully with the further increase of the spatio-temporal resolution of EUV imaging, such as

when the SolO approaches the closest perihelion. As driven by constant photospheric turbulent flows, it is reasonable to conjecture that the reconnection at smaller-scale null-points possibly occurs ubiquitously so as to heat the low corona. Assuming all fluxes are dissipated through this type of null-point reconnection driven by cancelling flux, based on the estimation given by ref. 35, the heating per unit area is $5 \times 10^6$ erg cm$^{-2}$ sec$^{-1}$ in the Quiet-Sun and is $5 \times 10^7$ erg cm$^{-2}$ sec$^{-1}$ in an active region, respectively, which are sufficient to heat the chromosphere. If 10–20% of this leaks to higher levels it would be sufficient for heating the low corona[70].

Except for the quasi-stable and persistent nature, the null-point reconnection also occurs impulsively but with a short-time period. Its coupling with the eruption of a mini-filament produced a spiral jet, which more quickly transferred mass and magnetic twist to the higher corona as interpreted in Fig. 7c. The more details revealed during the dynamic reconnection phase support recent observations[57] and 3D MHD modelling of spiral jets[56,71], which suggested that a slowly rising mini-flux rope reconnects with the inclined overlying field near a null-point, which may collapse into a breakout current sheet during the eruption[45,71,72], and the helical flux is released to the overlying field as a spiral jet. A careful inspection found the appearance of continuous BPs and a co-spatial mini-filament in Hα images, which provides a solid evidence for the existence of a mini-magnetic flux rope that is often presupposed in previous studies.

## Methods

### Instruments and data

We mainly used the EUI/HRI$_{EUV}$ 174 Å level 2 data (https://doi.org/10.24414/2qfw-tr95). The dataset are also part of SolO Observing Plan for studying the origin of slow solar wind. The field of view (FOV) of the EUI/HRI$_{EUV}$ is -16.8′ × 16.8′ with a pixel size of 0.492″, which corresponds to a linear scale of about 195 km at this distance. The remaining spacecraft jitter was removed by a cross-correlation technique[73]. The HRI$_{EUV}$ mainly focused on NOAA active region (AR) 12957 as shown in Fig. 1a, b. It performed observations from 09:40 UT to 10:40 UT, with a cadence of 5 s and obtained 720 frames in total. The contribution of the HRI$_{EUV}$ 174 Å emission, at the peak of the thermal response, is primarily due to Fe IX (at 171.1 Å) and Fe X (at 174.5 Å and 177.2 Å). The instrument is sensitive to the emission from plasma at temperatures of roughly 1 MK.

The AIA and HMI are both on board Solar Dynamics Observatory (SDO)[74]. The pixel size and cadence of the AIA are 0.6″ (linear scale of 420 km) and 12 s, respectively. The calibrated HMI images have the same pixel size as the AIA images, but the cadence is 45 s and 12 min for the LOS and vector magnetograms, respectively. In addition, to explore the response of the null reconnection at transition region and chromospheric temperatures, we also took advantage of 1330 Å slit-jaw images from the IRIS, which has a resolution of 0.33″ (linear scale of 230 km) and cadence of 2 s, as well as the Hα images observed by the NVST, with a cadence of 12 s. Note that, the separation angle between SDO and SolO was ~6.7° on 3 March 2022.

### DEM inversion and uncertainty

The DEM is reconstructed by the "xrt_dem_iterative2.pro" routine in the Solar Software (SSW) based on six co-aligned AIA EUV passbands (131 Å, 94 Å, 211 Å, 335 Å, 171 Å, and 193 Å). The temperature range of inversion is set as $5.5 \leq \log T \leq 7.5$, following with previous suggestions[75–77]. The DEM-weighted average temperature and total emission measure (EM) are derived from the following two equations:

$$\bar{T} = \int DEM(T) \times T dT / \int DEM(T) dT \tag{1}$$

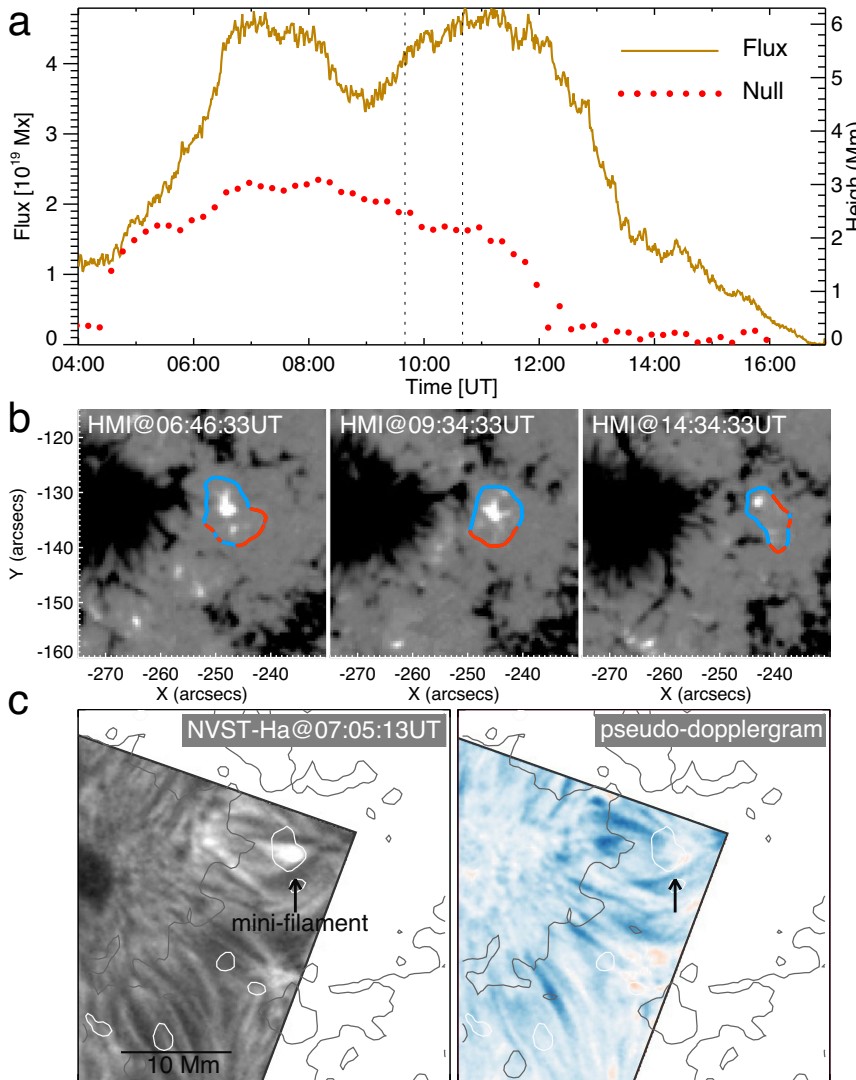

**Fig. 6 | Driver of null-point reconnection. a** Temporal evolutions of the total magnetic flux of the minor positive polarity generating the null-point configuration (solid line) and the height of the null-point (dotted line). Two vertical dashed lines represent the start and end times of the HRI$_{EUV}$ observation. **b** Radial components of the HMI vector magnetograms at three times. The curves in blue show the PILs with the segments in red indicating the locations of BPs. **c** NVST Hα intensity map and corresponding pseudo-Doppler map overlaid by the contours (±100G) of the HMI LOS magnetogram with the positive (negative) in white (grey). The arrow points out the mini-filament. The Hα pseudo-Doppler image, derived by subtracting the Hα red wing image (at +0.4 Å) from the Hα blue wing image (at −0.4 Å), only shows the direction of Doppler velocity.

and

$$EM = \int DEM(T)dT. \tag{2}$$

We have also tested the reliability of the DEMs through 200 Monte Carlo (MC) simulations, which are done by adding a random noise to observed intensities and rerunning the procedure again. Supplementary Fig. 1 plots the DEM result for the point-like brightening near the null-point, which shows that the emission originates from a wide temperature range ($5.7 \leq \log T \leq 7.4$), where the best-fit DEM solution is well constrained. The high-temperature component of the DEM is also indicated by the fact that the X-ray emission appears at the location of the point-like brightening as proved by the XRT images, although which are only available after 11:30 UT and have a very low resolution. We further calculate the DEM for an erupting blob as shown in Supplementary Fig. 2, which displays that the result is well constrained in the temperature range of $5.5 \leq \log T \leq 6.9$. The DEM-weighted average temperature is ~2 MK. The density $n$ is estimated by

$$n = \sqrt{\frac{EM}{l}}, \tag{3}$$

where $l$ is the depth of the blob along the LOS. Assuming the depth of the blob approximates its width (about 0.5 Mm), the density is calculated to be ~$5 \times 10^9$ cm$^{-3}$.

**Thermal energy estimation**
The thermal energy flow $\langle F \rangle$ caused by the reconnection outflow blobs is estimated by

$$\langle F \rangle = \frac{e v_b t_b f_b S_b}{S_n} \tag{4}$$

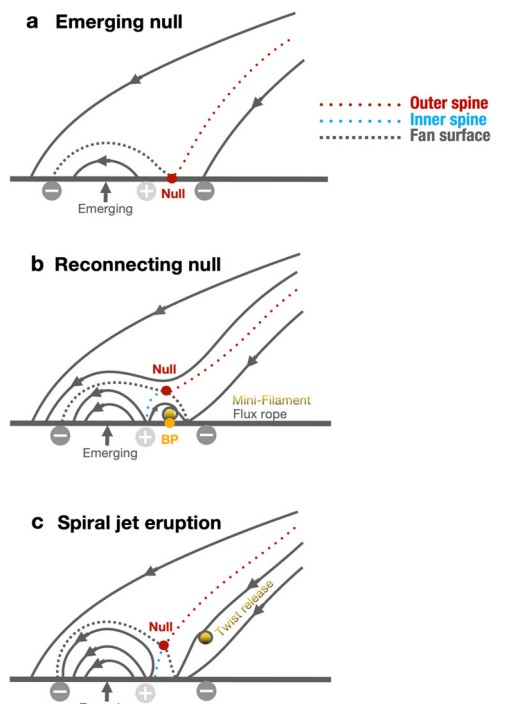

**a  Emerging null**

- ······ Outer spine
- ······ Inner spine
- ······ Fan surface

Emerging  Null

**b  Reconnecting null**

Null

Mini-Filament
Flux rope

Emerging  BP

**c  Spiral jet eruption**

Null

Twist release

Emerging

**Fig. 7 | Schematic drawing for the evolution of the null-point.** Magnetic field lines in a vertical plane through the null-point from the emergence (**a**) to reconnection phase (**b**), as well as the eruption of the mini-filament and induced spiral jet (**c**). The null-point and bald patch are denoted by dots in red and orange, respectively. The dotted lines in red and blue represent the outer and inner spines, respectively. The dashed separatrix field lines denote fan surface in 3D. The dots in gold indicate the twisted flux and mini-filament.

and the enthalpy density $e$ is

$$e = \frac{\gamma n_e k_B T_e}{\gamma - 1},\tag{5}$$

where $v_b$ and $t_b$ are the velocity and lifetime of reconnection blobs, respectively; $f_b$ is the frequency of blobs; $S_b$ is the projected area of blobs; $S_n$ denotes the area in which we can observe one null-point. Assuming $S_n$ approximates the whole area of the null-point configuration as observed here, it is about $(10\,\mathrm{Mm})^2$. Within the EUI observation window of 1 hour, 60 blobs are detected at least, thus $f_b$ is about $60\,\mathrm{h^{-1}} = 1.7 \times 10^{-2}\,\mathrm{s^{-1}}$. Then taking $T_e = 2\,\mathrm{MK}$, $n_e = 5 \times 10^9\,\mathrm{cm^{-3}}$, $v_b = 200\,\mathrm{km\,s^{-1}}$, $t_b = 40\,\mathrm{s}$, $S_b = (0.5\,\mathrm{Mm})^2$, the internal energy flow $\langle F \rangle$ is estimated to be $-1.2 \times 10^5\,\mathrm{erg \cdot cm^{-2} \cdot s^{-1}}$, which accounts for ~40% of the total energy flow required for heating the quiet-Sun corona. In comparison, the kinematic energy is negligible after estimation following the same procedure.

### 3D magnetic field and topology computation

We calculate the 3D coronal magnetic field from a potential field extrapolation based on the radial component of the HMI vector field as the bottom boundary. The locations of null-point and fan-spine are computed by the method in ref. 78. In the current case, considering the lack of observations of electric currents for the small-scale region of interest, we only take advantage of the potential field model. Overall, the potential field method is good enough given the cospatiality between the observed point-like brightening and extrapolated null-point and the agreement between the whole dome-like structure and the extrapolated fan surface. In fact, the null-point is a topologically rather-stable feature as confirmed by its long-term existence in the

current event (Fig. 6a). It is also shown that the appearance of observed current concentrations below the fan surface cannot destroy the null-point[46,79], usually only giving rise to a displacement of the null-point position and a change of curvature of the inner and outer spine lines as indicated by the slight deviation of the computed outer spine from the observed one (Fig. 3b). We further examine the influence of the bottom boundary used for extrapolation on the properties of the null-point. It is found that the height of the null-point systematically decreases by 1–2 Mm if the entire sunspot near the minor positive polarity is included.

The squashing degree Q, which measures the mapping of the field lines[80], is calculated by the method developed by ref. 81. Taking advantage of the HMI vector magnetograms, the BPs are calculated by using the formula

$$\mathbf{B_h} \cdot \nabla_h B_z|_{PIL} > 0,\tag{6}$$

where $\mathbf{B_h}$ and $B_z$ are the horizontal and vertical components of the vector magnetic field $\mathbf{B}$, respectively. In addition, we also calculate BPs through the bottom boundary of extrapolated 3D potential field data. It is found that the BPs appearing in the observed vector field is absent in the extrapolated one, strongly indicating the existence of a twisted flux rope.

### Data availability

The datasets generated during and/or analysed during the current study are available from the corresponding author upon request. The EUI data are also available at https://doi.org/10.24414/2qfw-tr95. The SDO and IRIS data are at http://jsoc.stanford.edu/ajax/exportdata2.html?ds=aia.lev1 and http://jsoc.stanford.edu/IRIS/IRIS.html, respectively. The NVST data are at http://fso.ynao.ac.cn/dataarchive.aspx. The XRT data are at http://solar.physics.montana.edu/HINODE/XRT/.

### Code availability

The codes for calculating DEM and 3D coronal magnetic field are available without any restrictions in Solar SoftWare (SSW) package at https://www.lmsal.com/solarsoft/. The codes for calculating the squashing degree Q are accessible from http://github.com/el2718/FastQSL.

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

## Acknowledgements

Solar Orbiter is a mission of international cooperation between ESA and NASA, operated by ESA. The EUI instrument was built by CSL, IAS, MPS, MSSL/UCL, PMOD/WRC, ROB, LCF/IO with funding from the Belgian Federal Science Policy Office (BELSPO/PRODEX PEA 4000134088, 4000112292, 4000117262, and 4000134474); the Centre National d'Etudes Spatiales (CNES); the UK Space Agency (UKSA); the Bundesministerium für Wirtschaft und Energie (BMWi) through the Deutsches Zentrum für Luft- und Raumfahrt (DLR); and the Swiss Space Office (SSO). AIA data are courtesy of NASA/SDO, a mission of NASA's Living With a Star Program. X.C., H.L., J.C., Y.L.W., M.D.D., Y.G., and W.T.F. are funded by NSFC grants 11722325, 11733003, 11790303, and 11790300, and by the National Key R&D Program of China under grant 2021YFA1600504. X.C. is also supported by the Alexander von Humboldt foundation. L.P.C. gratefully acknowledges funding by the European Union (ERC, ORIGIN, 101039844). G.A. acknowledges financial support from the French national space agency (CNES), as well as from the Programme National Soleil Terre (PNST) of the CNRS/INSU also co-funded by CNES and CEA. X.S. acknowledges financial support by National Key R&D Program of China (2021YFA1600503), NSFC grant 11790301, and the mobility program (M-0068) of the Sino-German Science Center. A.N.Z. thanks the Belgian Federal Science Policy Office (BELSPO) for the provision of financial support in the framework of the PRODEX Programme of the European Space Agency (ESA) under contract number 4000136424. D.M.L. is grateful to the Science Technology and Facilities Council for the award of an Ernest Rutherford Fellowship (ST/R003246/1).

## Author contributions

X.C. led the project, analysed observational data, and wrote the manuscript. H.L. extrapolated 3D coronal magnetic field and calculated magnetic topology skeleton with the guide of J.C. Y.W. estimated the released energy. E.P., G.A., L.C., H.P., X.Z., C.X., M.D., S.S., D.B., A.Z., G.Y. & L.H. discussed the results, and made revisions to the manuscript. L.C., H.P., D.B., L.T., R.C., A.Z., D.L., L.H., P.S., L.R., C.V., K.B., and S.P. prepared the EUI data.

## Competing interests

The authors declare no competing interests.
