## [Peer Review File · Nature Communications]

REVIEWER COMMENTS

Reviewer #1 (Remarks to the Author):

The authors have again gone to great lengths to address all my comments. The analysis is OK, but still my main concern is the following one.

The authors claim they are reporting on two breakthrough results:

1 - the scale of nullpoint reconnection we observed is the smallest ever observed in the corona;

2 - the null-point reconnection is found to proceed persistently, with a higher frequency than what reported previously.

I still think that both results, though very interesting, are of incremental nature, due to the availability of an instrument with higher resolution and cadence than before. I leave

it to the Editor to decide whether this is acceptable for Nature Communications.

Reviewer #2 (Remarks to the Author):

Ultra-high-resolution Observations of Persistent Null-point Reconnection in the Solar Corona

By X. Cheng et al.

In this third version of the originally submitted manuscript, the authors have answered all my comments and concerns, as well as included my suggested corrections. I appreciate the change of title and I find the present version suitable for publication (see a few comments at the end).

From my point of view, this manuscript shows how the high temporal and spatial resolution of new observations (EUI data) can improve our vision/understanding of similar phenomena discussed and analyzed in earlier studies using lower resolution data (magnetic reconnection at null points associated to different energy level events, origin and evolution of jets, presence of BPs associated with the formation of mini-filaments).

A few additional comments:

1) There are several typos in the text. Please, check.

2) "Such twisted field often gives rise to filaments or mini-filaments at their dips and represent locations where much energy is stored."

Why "much energy" and where is it stored? This need to be rephrased.

3) "Moreover, the mini-filament also displays a blueshift feature (the right panel of Figure 6c), which suggests that the mini-filament was ascending in height and then erupted to cause the jet once destabilization."

Do you mean "to cause the jet once destabilized"?

Reviewer #3 (Remarks to the Author):

I disagree with Cheng et al. responses. Their replies do not make any sense. As I mentioned in the previous reports, this paper does not add anything to the existing knowledge on the physics of jets. I have provided enough evidence (published observations and 3D MHD models) that this type of quasiperiodic reconnection (even better than reported here) in the small null-point topologies (so-called breakout reconnection) is very well observed before.

Therefore, I cannot recommend this paper to Nature communication.

REVIEWER COMMENTS

Reviewer #1 (Remarks to the Author):

The authors have again gone to great lengths to address all my comments. The analysis is OK, but still my main concern is the following one.

The authors claim they are reporting on two breakthrough results:

1 - the scale of nullpoint reconnection we observed is the smallest ever observed in the corona;

2 - the null-point reconnection is found to proceed persistently, with a higher frequency than what reported previously.

I still think that both results, though very interesting, are of incremental nature, due to the availability of an instrument with higher resolution and cadence than before. I leave it to the Editor to decide whether this is acceptable for Nature Communications.

Reply: We very appreciate your constructive comments during the three rounds of peer review processes. We agree with you that the availability of the instrument with higher resolution and cadence enables our findings of magnetic reconnection occurring persistently at a minor null-point. In previous studies, we acknowledge that the null-point configuration either for large-scale flares or small-scale jets has been derived, however, the lower spatial-temporal resolution data such as from SDO/AIA prevent from determining the precise null-point location and the concrete relationship between the null-point reconnection and induced dynamic features. That is to say, whether the reconnection precisely takes place at a null-point is often ambiguous observationally. With the picture constructed through the theory and MHD simulations, previous observational studies well interpreted many related phenomena, but which does not mean that the theoretical and MHD results are the final ground truth. We need better data to verify the present understanding of the null-point reconnection and rethink its meanings.

Taking the papers (Kumar et al. 2018, 2019, 2022) the third referee mentioned as examples, they interpreted many jet events using the MHD modelling developed by Wyper et al., but some questions remain. In their observations, the EUV brightenings used for justifying null-point reconnection are unknown to be located at the null-point or related fan surface. Here, the co-spatiality between the calculated null-point and point-like brightening confirms the reconnection occurring at the null-point. Moreover, they only detected unidirectional blobs, but whose birthplaces cannot be identified. Here, we observed bidirectional blobs that not only moved along the outer spine but also along the fan surface, thus clearly showing that they were from the null-point.

We admit that our results are more about the confirmation of the occurrence and persistence of null-point reconnection, rather than fundamental physics of null-point reconnection on microscales, which is impossible to be settled by remote-

sensing imaging observations. However, we believe that our results, more importantly, present a promising path for solving the coronal heating problem. At page 15 of the revised manuscript, we reinforced the discussions about applications of our results. The modified sentences are: “The finding of persistent minor null-point reconnection sheds a new light on the solution of the coronal heating problem. As revealed by SUNRISE II, minor small-scale opposite-polarity fluxes are prevalent at the periphery of the penumbra in the moat around a sunspot[34]. In quiet-Sun regions, opposite-polarity fluxes frequently appear within dominant flux concentrations although they may be short-lived[63, 64]. We even tentatively calculated the topology of the potential field over a larger quiet-Sun region nearby the null-point studied currently and found abundant low-lying small-scale null-points (see Figure 1 shown below). Our observations thus tend to support the discovery of even smaller and more frequent null-point reconnection events, in particular over the quiet-Sun region, hopefully with the further increase of the spatio-temporal resolution of EUV imaging, such as when the SoI approaches the closest perihelion. As driven by constant photospheric turbulent flows, it is reasonable to conjecture that the reconnection at smaller-scale null-points possibly occurs ubiquitously so as to heat the low corona.”

We hope that these updates can help alleviate your main concern for our manuscript.

Figure 1. Left: HMI radial magnetic field distribution at a larger region including the target active region. Right: The topology structures of 3D potential field above the quiet-Sun region that is indicated by the black box in the left panel. Pluses in red show the locations of null-points, the curves in yellow represent the outer spines, the lines in cyan are the field lines related to fan surfaces.

Reviewer #2 (Remarks to the Author):

Ultra-high-resolution Observations of Persistent Null-point Reconnection in the Solar Corona By X. Cheng et al.

In this third version of the originally submitted manuscript, the authors have answered all my comments and concerns, as well as included my suggested corrections. I appreciate the change of title and I find the present version suitable for publication (see a few comments at the end).

From my point of view, this manuscript shows how the high temporal and spatial resolution of new observations (EUI data) can improve our vision/understanding of similar phenomena discussed and analyzed in earlier studies using lower resolution data (magnetic reconnection at null points associated to different energy level events, origin and evolution of jets, presence of BPs associated with the formation of mini-filaments).

Reply: We are grateful that our new manuscript has eliminated your concerns satisfactorily.

A few additional comments:

1) There are several typos in the text. Please, check.

2) "Such twisted field often gives rise to filaments or mini-filaments at their dips and represent locations where much energy is stored."

Why "much energy" and where is it stored? This need to be rephrased.

3) "Moreover, the mini-filament also displays a blueshift feature (the right panel of Figure 6c), which suggests that the mini-filament was ascending in height and then erupted to cause the jet once destabilization." Do you mean "to cause the jet once destabilized"?

Reply: Your new comments have been incorporated into the revised manuscript.

Reviewer #3 (Remarks to the Author):

I disagree with Cheng et al. responses. Their replies do not make any sense. As I mentioned in the previous reports, this paper does not add anything to the existing knowledge on the physics of jets. I have provided enough evidence (published observations and 3D MHD models) that this type of quasiperiodic reconnection (even better than reported here) in the small null-point topologies (so-called breakout reconnection) is very well observed before. Therefore, I cannot recommend this paper to Nature communication.

Reply: Thanks for your comment again. However, we would like to address that the key point of our manuscript is on observations of persistent magnetic reconnection at a minor null-point, the spiral jet is just one episode of the small-scale reconnection process. In order to further clarify advances and differences of our results to previous ones, we made modifications in the new manuscript as follows:

At the first paragraph of page 10, we stated "These processes with more details as revealed here confirm the scenario proposed for blowout jets caused by mini-filament eruptions in previous lower resolution data[51–54]."

At the beginning of the second paragraph, "Compared with previous studies, the current observations show a new characteristic, i.e., the heating and lateral propagation of the mini-filament leading front when interacting with the fan surface."

At the end of the second paragraph, “In addition, no evidence was found to support that the preceding null-point reconnection plays a role in triggering the mini-filament eruption and causes the spiral jet as argued previously [43, 44, 66, 67] because the null-point reconnection remained sustained after the eruption (Figure 4a and 4b).”

We also reorganized the last paragraph of page 16, the new one is “Except for the quasi-stable and persistent nature, the null-point reconnection also occurs impulsively but with a short-time period. Its coupling with the eruption of a mini-filament produced a spiral jet, which more quickly transferred mass and magnetic twist to the higher corona as interpreted in Figure 7c. The more details revealed during the dynamic reconnection phase support recent observations[66] and 3D MHD modelling of spiral jets[67, 68], which suggested that a slowly rising mini-flux rope reconnects with the inclined overlying field near a null-point, which may collapse into a breakout current sheet during the eruption[45, 68, 69], and the helical flux is released to the overlying field as a spiral jet. A careful inspection found the appearance of continuous BPs and a co-spatial mini-filament in Ha, which provides a solid evidence for the existence of a mini-magnetic flux rope which is often presupposed in previous studies.”

REVIEWERS' COMMENTS

Reviewer #2 (Remarks to the Author):

Ultra-high-resolution Observations of Persistent Null-point Reconnection in the Solar Corona

By X. Cheng et al.

I thank the authors for modifying the text following the minor comments in my third report. For this fourth version I will repeat what I said previously:

“From my point of view, this manuscript shows how the high temporal and spatial resolution of new observations (EUI data) can improve our vision/understanding of similar phenomena discussed and analyzed in earlier studies using lower resolution data (magnetic reconnection at null points associated to different energy level events, origin and evolution of jets, presence of BPs associated with the formation of mini-filaments).”

However, I have a comment related to the authors' answer to the first reviewer. It is “clear/highly probable” (use the words you prefer) that the lower you go in the solar atmosphere, i.e. the closer you are to the photosphere in the quiet Sun, more null points will be found even after taking into account the noise in the magnetic field measurements. This is a consequence of the salt-and-pepper nature of the photospheric field evident in quiet Sun regions. The number of null points, associated fans and spines, will increase because the chance to have polarities of one sign surrounded by the opposite sign polarity will increase (as they show in the figures included in their answer). I hope I am clear.

Furthermore, Longcope & Parnell (2009) and Longcope, Parnell, De Forest (2009), the first using MDI and the second comparing Hinode's NFI and MDI, have estimated the variation of the coronal null point density with height. These results are directly related to what the authors have added in their new version of the manuscript and I suggest to add a reference to these works. Furthermore, Schrijver & Title (2002) find similar results using a simulation and comparing to TRACE and SOHO/MDI; they also discuss the inference of their results for the heating of the quiet-Sun corona.

My comment is not disqualifying the authors' addition of the new text and its implications for (e.g.) coronal heating.

I do expect that higher spatial and temporal resolution observations will support and improve previous results/findings, as I say in my general comment on this manuscript.

\bibitem[Longcope et al.(2009)]{2009ASPC..415..178L} Longcope, D., Parnell, C., \& DeForest, C.\ 2009, The Second Hinode Science Meeting: Beyond Discovery-Toward Understanding, 415, 178. doi:10.48550/arXiv.0901.0865

\bibitem[Longcope \& Parnell(2009)]{2009SoPh..254...51L} Longcope, D.~W. \& Parnell, C.~E.\ 2009, \solphys, 254, 51. doi:10.1007/s11207-008-9281-x

\bibitem[Schrijver and Title(2002)]{2002SoPh..207..223S} Schrijver, C.~J., Title, A.~M.\ 2002.\ The topology of a mixed-polarity potential field, and inferences for the heating of the quiet solar corona.\ Solar Physics 207, 223–240. doi:10.1023/A:1016295516408

REVIEWER COMMENTS

Reviewer #2 (Remarks to the Author):

Ultra-high-resolution Observations of Persistent Null-point Reconnection in the Solar Corona By X. Cheng et al.

I thank the authors for modifying the text following the minor comments in my third report. For this fourth version I will repeat what I said previously: “From my point of view, this manuscript shows how the high temporal and spatial resolution of new observations (EUI data) can improve our vision/understanding of similar phenomena discussed and analyzed in earlier studies using lower resolution data (magnetic reconnection at null points associated to different energy level events, origin and evolution of jets, presence of BPs associated with the formation of mini-filaments).”

However, I have a comment related to the authors’ answer to the first reviewer. It is “clear/highly probable” (use the words you prefer) that the lower you go in the solar atmosphere, i.e. the closer you are to the photosphere in the quiet Sun, more null points will be found even after taking into account the noise in the magnetic field measurements. This is a consequence of the salt-and-pepper nature of the photospheric field evident in quiet Sun regions. The number of null points, associated fans and spines, will increase because the chance to have polarities of one sign surrounded by the opposite sign polarity will increase (as they show in the figures included in their answer). I hope I am clear. Furthermore, Longcope & Parnell (2009) and Longcope, Parnell, De Forest (2009), the first using MDI and the second comparing Hinode’s NFI and MDI, have estimated the variation of the coronal null point density with height. These results are directly related to what the authors have added in their new version of the manuscript and I suggest to add a reference to these works. Furthermore, Schrijver & Title (2002) find similar results using a simulation and comparing to TRACE and SOHO/MDI; they also discuss the inference of their results for the heating of the quiet-Sun corona. My comment is not disqualifying the authors’ addition of the new text and its implications for (e.g.) coronal heating. I do expect that higher spatial and temporal resolution observations will support and improve previous results/findings, as I say in my general comment on this manuscript.

\bibitem[Longcope et al.(2009)]{2009ASPC..415..178L} Longcope, D., Parnell, C., \& DeForest, C.\ 2009, The Second Hinode Science Meeting: Beyond Discovery-Toward Understanding, 415, 178. doi:10.48550/arXiv.0901.0865

\bibitem[Longcope \& Parnell(2009)]{2009SoPh..254...51L} Longcope, D.-W. \& Parnell, C.-E.\ 2009, \solphys, 254, 51. doi:10.1007/s11207-008-9281-x

\bibitem[Schrijver and Title(2002)]{2002SoPh..207..223S} Schrijver, C.-J., Title, A.-M.\ 2002.\ The topology of a mixed-polarity potential field, and inferences for the heating of the quiet solar corona.\ Solar Physics 207, 223–240. doi:10.1023/A:1016295516408

Reply: Thanks for your further comments, we have addressed your new point and added the references you recommended in pages 14-16 of the new manuscript.